# Post-Translational Modifications of PCNA: Guiding for the Best DNA Damage Tolerance Choice

**DOI:** 10.3390/jof8060621

**Published:** 2022-06-10

**Authors:** Gemma Bellí, Neus Colomina, Laia Castells-Roca, Neus P. Lorite

**Affiliations:** Departament de Ciències Mèdiques Bàsiques, Institut de Recerca Biomèdica de Lleida, Universitat de Lleida, 25198 Lleida, Spain; laia.roca@gmail.com (L.C.-R.); neus.perez@udl.cat (N.P.L.)

**Keywords:** PCNA, DNA damage tolerance, DNA replication stress, fungal genome stability, DNA replication forks, post-translational modifications, translesion synthesis, template switch, salvage recombination

## Abstract

The sliding clamp PCNA is a multifunctional homotrimer mainly linked to DNA replication. During this process, cells must ensure an accurate and complete genome replication when constantly challenged by the presence of DNA lesions. Post-translational modifications of PCNA play a crucial role in channeling DNA damage tolerance (DDT) and repair mechanisms to bypass unrepaired lesions and promote optimal fork replication restart. PCNA ubiquitination processes trigger the following two main DDT sub-pathways: Rad6/Rad18-dependent PCNA monoubiquitination and Ubc13-Mms2/Rad5-mediated PCNA polyubiquitination, promoting error-prone translation synthesis (TLS) or error-free template switch (TS) pathways, respectively. However, the fork protection mechanism leading to TS during fork reversal is still poorly understood. In contrast, PCNA sumoylation impedes the homologous recombination (HR)-mediated salvage recombination (SR) repair pathway. Focusing on *Saccharomyces cerevisiae* budding yeast, we summarized PCNA related-DDT and repair mechanisms that coordinately sustain genome stability and cell survival. In addition, we compared PCNA sequences from various fungal pathogens, considering recent advances in structural features. Importantly, the identification of PCNA epitopes may lead to potential fungal targets for antifungal drug development.

## 1. Introduction. Proliferating Cell Nuclear Antigen (PCNA)

The sliding clamp proliferating cell nuclear antigen (PCNA), encoded by POL30 in *S. cerevisiae* [1], is central to DNA replication and repair [2,3], and other fundamental processes of DNA metabolism, such as chromatin assembly [4,5,6], sister chromatid cohesion establishment [7,8,9], lagging strand maturation [10,11], epigenetic conversions and inheritance [12,13,14], gene expression [15,16] and DNA damage checkpoint pathway activation [17]. Initially, PCNA was discovered as a biological marker for systemic lupus erythematosus disease [18]. In the early eighties, PCNA was identified as a cyclin [19,20] due to its role in proliferation and correlative expression with the cell cycle. Subsequently, it was characterized as a processivity factor for replicative DNA polymerase (Pol) δ [21,22,23], Polα and β [24], and Polε [25,26]. Therefore, PCNA was related to DNA replication [27] and it was given essential roles on the replication fork (RF), including leading- and lagging-strand synthesis coordination in combination with replication factor C (RFC) [24], and the synthesis of the leading strand during the elongation stage [28]. PCNA was soon associated with nucleotide excision repair [29,30], together with the role of DNA Polδ in UV-induced DNA damage [31,32]. Concomitantly, the function PCNA in DNA damage repair was elucidated [33,34,35]. Indeed, PCNA has become a major scaffold protein for DNA damage response by participating in multiple DNA repair pathways, including DNA mismatch repair [36,37,38], base excision repair (BER) [39,40,41,42], double-strand break repair [43,44,45], translesion synthesis (TLS) [46,47,48,49,50], the resolution of deleterious interactions between replication and transcription machinery [51], and in the post-translational regulation of DNA damage tolerance (DDT) through a selection of various sub-pathways of the post-replication repair pathway [52].

In this review, we briefly present the basic features of PCNA during the unperturbed cell cycle, focusing on *S. cerevisiae* budding yeast cells. We summarize how post-translational modifications of PCNA are critical to channeling specific molecular pathways in response to stalled RFs. Current knowledge concerning DDT pathways regulation is highlighted. Furthermore, we provide a comparative study based on multiple alignments of PCNA amino acid residues from various fungal pathogens causing systemic infections. Specific differences in fungal PCNA epitopes reveal this essential protein as a potential therapeutic target.

## 2. PCNA Structural Features

PCNA belongs to the conserved family of DNA sliding clamps (β clamps) [53,54,55,56]. It forms a homotrimer in a closed head-to-tail ring structure around DNA. Each PCNA molecule has two globular domains linked by the inter-domain connecting loop (IDCL) [56]. The PCNA homotrimer is disposed in a pseudo-six-fold symmetry structure with an external layer of six β-sheets and an internal layer of twelve α-helices. The negatively charged outer surface potentially inhibits unspecific protein interactions, while the positively charged inner surface of the clamp structure faces the DNA duplex [53,54,55]. This interaction occurs through five basic lysine residues sliding on successive phosphates of one DNA strand by a spiral movement [57,58]. Since DNA is a helical molecule, the sliding of the PCNA homotrimer is tilted 30° and moves forward to backward every one-half turn of DNA in a process defined as “cogwheeling” [58]. The clamp tilt at the 3′ end of the DNA molecule may specify PCNA binding partners [59]. Another proposed mechanism for sliding is the “switch” from five of the twelve α-helices tracked by the DNA, which consequently changes the relative tilt to the 3′ terminus [57]. There exists a dynamic association/dissociation process, in which RFC loads PCNA onto RFs for its interaction with replicative Polδ [60]. In *S. cerevisiae*, PCNA only interacts with Pol3 (the catalytic subunit of Polδ) providing enough room between the inner walls of PCNA and the DNA to hold water molecules, which may facilitate sliding [61]. Once the polymerase dissociates, PCNA diffuses away from the primer terminus. Later, it re-associates with the polymerase to finish replication. This back-and-forward clamp motion has been suggested as the possible Polδ proofreading activity mechanism [58] or even the switch allowing translesion DNA polymerase to bypass DNA lesions [59]. Additionally, post-translational modifications also occur on the surface of the PCNA clamp to regulate DNA sliding [62,63] and in response to DNA damage [64].

PCNA functions as a platform for a large amount of DNA replication and editing enzymes [65,66]. Many of the PCNA interactors exhibit a short linear motif (SLiM) [67]. The PCNA binding motif is known as the PCNA-interacting peptide (PIP) box [68], through which many PCNA-interacting proteins are recruited to function in DNA replication [69]. Its extended PIP degron version targets PCNA for degradation [70]. A second important interaction motif is the AlkB homologue 2 PCNA interacting motif (APIM) [71,72,73], which mediates the interaction with genotoxic responsive proteins [69]. Both motifs are topologically similar and are localized in the hydrophobic pocket formed in the IDCL region of the β-sheets external surface of the ring [74,75]. Moreover, another PCNA interaction motif named the KA box has been described [76,77]. PCNA homotrimer may interact with three different partners synchronically through the three IDCL regions, facilitating spatio-temporal coordination for a multiplicity of purposes. Besides, the PIP box may overlap with the Rev1-interacting region (RIR)- and the Mlh-1 interacting protein (MIP) motifs [78], both of which include proteins related to DNA repair, enhancing the number of interactions with genome maintenance interactors (see details in the text). Interestingly, several PCNA binding partners are considered intrinsically disordered proteins (IDPs) or hold intrinsically disordered regions (IDRs) [79,80]. IDPs are unstructured in physiological conditions and fold to stable structures when binding their functional targets [81]. Rather abundant in eukaryotes, IDPs are relevant members of cellular signaling and become highly regulated at the post-translational level [82]. Therefore, the existence of three hydrophobic binding pockets in PCNA clamps, the redundancy of SLiM sequences, and the numerous IDPs interactors increase the complexity of the regulation of PCNA-mediated processes.

### 2.1. Loading and Unloading PCNA onto Duplex DNA

Cellular nuclei contain both homotrimeric free PCNA and the PCNA-DNA complex, which dynamically converge with the help of the loading and unloading machinery in a cyclic manner [83]. Clamp loaders belong to the AAA+ ATPases, and generate the mechanical force [84] to open and load PCNA homotrimers onto the single-strand DNA/double-strand DNA junction of the RF. Once the PCNA function is finished, it is unloaded from the DNA by an unloading complex. The loading/unloading cycle must be tightly regulated for efficient genomic replication and repair [85,86]. PCNA is loaded onto DNA by the RFC complex, which consists of RFC1 (large subunit) and RFC2, 3, 4 and 5 (small subunits). Other members of the same family are the RFC-like complexes (RLCs), which also contribute to the regulation of the chromatin-association of DNA clamps [87,88]. *S. cerevisiae* harbors the following three RLCs: Ctf18, Elg1 and Rad24 (CTF18, ATAD5 and RAD17, respectively, in humans), as large subunits and the following four small subunits: RFC2, 3, 4 and 5 [89].

RFC loads PCNA on primer-template junctions. The spiral-shaped RFC complex assembles with PCNA and opens the PCNA ring at the expense of ATP. After DNA binding, the PCNA clamp closes and the RFC dissociates [90,91,92]. Yeast and human Ctf18-RLC can load PCNA on gapped DNA, but in a less efficient manner than the RFC complex; however, Rad24-RLC and Egl1-RLC cannot [93,94]. Genome-wide PCNA occupation experiments showed that the RFC complex primarily loads PCNA on the lagging strand, while Ctf18-RLC preferentially loads PCNA on the leading strand [8]. Upon checkpoint activation, the Rad24-RFC clamp loader transports the Rad17p-Mec3p-Ddc1p complex to DNA lesions.

Once chromosomal DNA is duplicated, PCNA unloads from the DNA. The ATAD5-RLC unloader (Elg1-RLC in yeast) dissociates PCNA from DNA after DNA replication and repair [85,94,95,96]. Different studies reinforce the Elg1-RFC complex as a PCNA unloader genome [97,98,99]. Indeed, the absence of Elg1, as well as defects in PCNA unloading, lead to chromosome instability [100,101,102]. Despite controversy on the loading and unloading role of the Ctf18-RFC clamp [93,94], PCNA unloading by the RFC2, 3, 4 and 5 and the RFC2 and 5 subcomplexes has been reported in vitro [103]. Additionally, PCNA unloading is regulated by distinct mechanisms, namely replisome and nucleosome recruitment [94,104,105,106,107], ubiquitination of PCNA [108,109], PCNA acetylation followed by degradation [64], or the eventual dissociation of PCNA from DNA independently of unloaders [103].

### 2.2. PCNA and Replicative DNA Polymerases: Leading and Lagging DNA

The B-family of DNA polymerases α, δ and ε replicate chromosomal DNA in eukaryotic cells. DNA polymerization occurs in the 5′ to 3′ direction in both antiparallel strands of DNA at the RF. Polα and the primase complex not only start the synthesis on the leading strand but also constantly produce primer–template junctions on the lagging strand. PCNA is loaded onto primer–template junctions, which Polδ uses to polymerize the DNA of the lagging strand. For leading strand synthesis, Polδ polymerizes the DNA of the first Okazaki fragment (over the replication origin), followed by Polε, which performs the continuous leading strand synthesis [110,111]. Polδ and Polε are among the most accurate DNA polymerases [112]. Conversely, Polα lacks 3′ to 5′ proofreading exonuclease activity. Notably, while Polε may only correct its own erroneously incorporated nucleotides, Polδ is able to replace both its own and Polε errors [113,114]. In fact, recent evidence has placed Polδ as primarily responding to the stalling of leading-strand synthesis by surpassing other polymerases [115]. The stalling of Pol *ε* causes the uncoupling of leading-strand synthesis from template unwinding. Nevertheless, template unwinding and lagging-strand synthesis continue, generating stretches of RPA-coated ssDNA on the leading strand. Switching replicative polymerases allows for the rapid continuation of replication after uncoupling, by ensuring high-fidelity replicative polymerase DNA synthesis on the leading strand, to elude mutagenic DDT processes and checkpoint activation [116] (Figure 1). Once DNA synthesis is finished, the complex regulation of PCNA unloading is tightly controlled by different processes [64,94,98,103,104,105,106,107,109,117]; then, PCNA clamps multiple enzymes necessary for chromosome assembly [4,5,6], sister chromatid cohesion [7,8,9] or gene expression [15,16].

## 3. Post-Translational Modifications of PCNA and the DDT Pathways

Endogenous processes, as well as exogenous DNA damaging agents, such as UV-irradiation and alkylating agents, may lead to the formation of bulky lesions that eventually stall RFs. Organisms have developed a collection of mechanisms to ensure the progression of DNA replication in the presence of DNA-disturbing events, the DDT pathways. The post-translational modification of PCNA triggers DNA damage response, mainly by determining the interacting proteins to be recruited. Three main DDT pathways can be involved, which are as follows: the translesion synthesis (TLS), template switch (TS) and salvage recombination (SR) pathways (Figure 2).

### 3.1. Translesion Synthesis (TLS)-Mediated DDT Error-Prone Pathway

Bulky lesions may prevent DNA synthesis by classical replicative DNA polymerases, consequently leading to stalled RFs. In this scenario, genome instability and chromosome rearrangements may compromise cell viability. To avoid prolonged uncoupling, cells trigger a DDT mechanism termed TLS that allows replication to proceed through damaged DNA or stalled RFs. However, replicative DNA polymerases are not efficient at incorporating nucleotides when the opposite DNA template is damaged. Instead, TLS is executed by specialized TLS polymerases, which are less stringent to DNA damage and are able to synthesize across DNA lesions, with no associated proofreading activity. The decrease in fidelity may trigger the accumulation of mutations. In fact, TLS represents the major accumulation of mutations by an error-prone bypass pathway in eukaryotes [118,119,120,121,122,123,124,125,126,127].

#### 3.1.1. TLS Polymerases Structural Features

TLS polymerases belong to the B-family (Polζ) and the Y-family (Polη, Polι, Polκ and Rev1) of DNA polymerases [128]. Despite being highly conserved throughout evolution [129], the TLS pathway displays substantial variability in polymerase distribution among species. Thus, only Polζ, Polη and Rev1 are present in budding yeast.

**-Polζ** comprises the Rev3 catalytic subunit and Rev7, Pol31 and Pol32 accessory subunits. Former in vitro studies determined that Rev3 physically interacts by its N-terminal region with Rev7, with both being subunits required for a minimally functional complex [130]. Later, it was shown that Pol31 and Pol32, which are both subunits of Polδ, were purified along with Rev3-Rev7 to form a fully functional complex [131,132,133,134]. The recently resolved structure of Polζ reveals the presence of a pentameric ring conformation that contains two Rev7 subunits, in addition to Rev3, Pol31 and Pol32 [135]. Similarly to other members of the B-family DNA polymerases, Rev3 harbors two conserved metal-binding motifs of cysteine, CysA and CysB (Figure 3a). The zinc finger (ZF) motif CysA is placed toward the N-terminal side of the CTD and mediates DNA-dependent interactions of Polζ with PCNA. The CysB motif containing a [4Fe-4S] cluster, located in the C-terminal portion of the CTD, serves as a docking site for additional polymerase subunits, such as Pol31 in yeast and POLD2 in mammalian cells. The substitution of cysteine residues that coordinate the [4Fe-4S] cluster prevents the specific binding of Rev3-Pol31 [130,136].

Disorder prediction algorithms show that Rev3, Rev7 and Pol31 are mostly structured. Conversely, Pol32, which is uniquely attached to the complex by Pol31 [137], has an IDR at the C-terminus containing a PIP motif that may bind PCNA [138]. Polζ plays a major role in extending mismatched primer termini in both spontaneous and damage-induced mutagenesis [139]. Thus, the deletion of Rev3 eliminates 50–70% of spontaneous mutations and more than 90% of damage-induced mutagenesis in *S. cerevisiae* cells [140,141,142].

**-Polη** is thought to be a first responder in TLS, being rapidly recruited to stalled RFs. It was first identified in yeast due to its ability to replicate UV light-induced DNA lesions such as cis-syn thymine-thymine (TT) and cyclobutane pyrimidine dimers (CPD), in an error-free manner [143,144]. In humans, Polη was first identified as the mutated product of the *XPV* gene in patients with the xeroderma pigmentosum-variant [145].

Polη presents an IDR at the C-terminus [138,146] with a small ubiquitin-binding zinc finger (UBZ) motif that may interact with the ubiquitin moiety on ubiquitin-modified PCNA [146]. It also contains a PIP motif, PIP1, which binds to a hydrophobic pocket on the front face of PCNA or to a hydrophobic pocket on the CTD of Rev1, in a mutually exclusive manner [147,148]. Recently, a second PIP motif was found, termed PIP2. Both PIP1 and PIP2 share the ability to mediate the interactions with Rad6-Rad18, with PIP1 displaying a higher affinity than PIP2. Multiple PIP motifs on Polη may facilitate the recruitment of Polη to the complex to optimize TLS [149] (Figure 3b). Biochemical studies reveal that PCNA-binding stimulates the efficiency of nucleotide incorporation opposite both undamaged and damaged sites, achieved primarily by a reduction in the *Km* for the nucleotide, without affecting its low processivity or fidelity [150].

**-Rev1** contains a polymerase domain, a polymerase-associated domain (PAD) that is the active site that coordinates the essential metallic ions required for the nucleotidyl transferase reaction, and two IDRs, whereby the N-terminal IDR encloses a BRCT (breast cancer-associated protein-1 C-terminal) domain that binds on the front of PCNA at a site that partially overlaps with the PIP motif-binding site [151,152], while the C-terminal IDR contains two small ubiquitin-binding motifs (UBM) [153], and a small CTD [72,154,155]. This four-helix bundle binds to a region of the catalytic core of Polζ [156,157]. In addition, it binds to the PIP motifs of Polη, at a distinct site from where Polζ binds [147]. Subsequently, Rev1 may simultaneously bind Polζ and Polη (Figure 3c).

Compared to replicative polymerases, TLS polymerases exhibit intrinsic structural features that couple with their role in synthesizing damaged DNA [116]. In this sense, the Y-family DNA polymerases hold an especially flexible conformation at active sites to provide room to tolerate a variety of bulky, damaged template bases [158,159]. Moreover, two of their domains, involved in choosing and positioning the correctly paired nucleotides in the active site of the polymerase, are shorter and make fewer contacts with the DNA and the incoming dNTP [139]. These domains clench DNA, consequently adopting a DNA–dNTP binding closed conformation that may alter the mechanism of the proper selection of dNTP [139,158,160,161]. Additionally, the ability to excise mismatched dNTPs disappears, since they do not display proofreading exonuclease activity. In the case of the B-family Polζ, its ZF domain moves from an open to a closed conformation after binding the proper dNTP, an ability that confers higher fidelity compared to the Y-family DNA polymerases [135]. Nevertheless, in Polδ, the efficient extension of a mismatch is prevented by contacting the terminal base pair with the linker between the NTD and the PAD. Conversely, in Polζ, these contacts are absent, triggering the inactivation of the exonuclease domain [123,162] (extensively reviewed in [154,159]).

#### 3.1.2. Monoubiquitin-PCNA Modification Mediated by Rad6-Rad18

The activation of TLS involves post-translational modifications of the sliding clamp PCNA, consisting of monoubiquitination at highly conserved lysine K164 by the Rad6 E2 ubiquitin conjugase and the Rad18 E3 ubiquitin ligase. Although PCNA monoubiquitination is an essential step in TLS, its specific role is still not known in detail. However, it has been assumed that ubiquitin-modified PCNA is a signal to recruit TLS polymerases to stalled RFs [124,153,163,164,165,166]. Accordingly, an increased activity of both Rev1 and Polη is observed when binding ubiquitin-modified PCNA, compared with unmodified PCNA [166].

Cells lacking *RAD6* are extremely sensitive to a large variety of DNA-damaging agents, such as UV and ionizing irradiation, among others [167,168,169]. Besides, cells display slow growth and impaired meiotic recombination and sporulation [170,171]. Rad6 E2 ubiquitin-conjugating activity is required to achieve all its known functions [172]. Furthermore, different E3 ubiquitin ligase enzymes are known to recruit Rad6 to the specific target. Thus, besides Rad18, E3-Bre1 is responsible for H2A and H2B ubiquitination [173], while E3-Ubr1 participates in the N-end rule pathway [174]. Nonetheless, PCNA’s DDT activity is exclusively accomplished through interaction with Rad18 [175]. Although less severe, *RAD18* mutants also exhibit sensitivity to DNA-damaging agents and growth defects. Unlike *RAD6* mutants, *∆rad18* mutant cells do not exhibit defects in meiotic recombination, sporulation or N-end rule protein degradation. Evolutionary conserved from yeast to mammals [176], Rad18 contains a C3HC4 zinc finger (ZF) domain termed RING (really interesting new gene) that is characteristic of E3 enzymes, a SUMO interacting motif (SIM) that promotes the sumoylation of Rad18 in vivo and in vitro, and a C2H2 ZF domain for nucleic acid binding. Contradictory results are found in the literature about the role of the Rad18 C2H2 ZF. Peptides containing the ZF of human Rad18 were shown to bind ubiquitin, both in vitro and in vivo [177]. In contrast, human Rad5 orthologs, SHPRH and HLTF, were both shown to bind human Rad18 via its ZF in a competitive manner with ubiquitin [178]. More recently, Frittmann et al. identified the ZF motif of Rad18 as the Rad5 binding domain [179]. Rad18 also contains a Rad6-binding domain, R6BD [180] and a SAP (SAF-A/B, Acinus, PIAS) domain that mediates its interaction with DNA [180] (Figure 3d). Both DNA-binding and nucleotide-binding activities may promote Rad18 recruitment to ssDNA at DNA lesions, in an ATP-dependent manner [181]. Human Rad18 has a Polη-binding domain at its C-terminus [180], which is not found in yeast. Interestingly, it has been shown that Rad6 forms a stable complex with Rad18 [182].

In *S. cerevisiae*, the persistent stalling of RFs generates an accumulation of Replication Protein A (RPA) that exhibits an exceptionally high affinity for ssDNA, quickly coating ssDNA templates downstream of stalled primer/template junctions [182], and forming extended filaments to protect ssDNA from degradation or from forming abnormal structures [126]; these filaments restrict PCNA to the upstream dsDNA region by avoiding its diffusion along ssDNA [183]. The Rad6-Rad18 complex is directly engaged with an RPA filament by the interactions between Rad18-RPA [159,180].

#### 3.1.3. DNA Polymerase Switching during TLS

The term polymerase switching refers to the process by which one DNA polymerase replaces a second one at the 3′-OH end of a primed DNA template. Two model strategies to switch replicative polymerases for TLS have been described. A first model is termed the PCNA “tool belt”, in which multiple binding proteins may be recruited to PCNA monomers that are not bound by Polδ. Thus, the ubiquitin moiety is located on the backside of PCNA, where TLS polymerases may be engaged, while Polδ remains bound at the front [184]. This model is supported by a recent work of the human Polδ-PCNA-FEN1 complex on DNA [73], suggesting this mechanism for flap cleavage in Okazaki fragment maturation. Lancey et al. [73] showed the ability of PCNA to adopt a 20° tilted position, leading to the destruction of the critical interactions for DNA synthesis between the catalytic subunit of Polδ and PCNA, while the polymerase remains bound to PCNA via the PIP box. Other pieces of evidence for the “tool belt” model of polymerase switching have been reported in other organisms such as *Escherichia coli* [185,186] and archaea [187].

The second proposed mechanism comprises the formation of Rev1 bridges, in which a TLS polymerase is linked to PCNA via Rev1, without directly interacting with the clamp. In this model, Polδ dissociates from PCNA and DNA to permit TLS. Rev1 interacts with PCNA through the BRCT domain, and through PIP-like motifs with other Y-family polymerases [78] or with the Rev7 subunit of Polζ [116,188]. Accordingly, numerous studies support a non-catalytic role of Rev1 in the recruitment of other TLS polymerases [189,190,191,192]. Single-molecule studies revealed that both mechanisms, “tool belt” and Rev1 bridges, are able to dynamically interchange without dissociation [116,147]. Furthermore, their relative contribution is likely to be lesion-specific (for more information on the subject, see [142]). Additionally, the fact that both the major DNA polymerase Polδ and the TLS Polζ require the same accessory subunits Pol31 and Pol32 [131,132,193] leads to the proposal of a possible switching mechanism between Polδ and Polζ. Pol3 dissociates from Pol31-Pol32 bound to PCNA at stalled forks, whereas the Rev3-Rev7 heterodimer is recruited by Pol31-Pol32-binding-PCNA [131,194].

Defective-Replisome-Induced-Mutagenesis (DRIM) occurs when problems in replication factors, affecting replisome integrity or Polα, Polδ or Polε, promote the use of Polζ to continue DNA synthesis copying undamaged DNA. As a consequence, the low fidelity of Polζ causes an increase in the mutational rate [195,196,197,198]. PCNA monoubiquitination at K164 and Polζ recruitment were also observed during DRIM. Moreover, hydroxyurea (HU), which impedes replication, induced Polζ DNA synthesis, independently of damage, suggesting that DRIM could function as the response to replication impediments [195]. These observations indicate that PCNA contributes to the increased mutagenesis observed in DRIM, which is independent of DNA damage. On the other hand, it has long been debated whether TLS occurs at the replication fork or, conversely, post-replicatively, behind the fork. In *S. cerevisiae*, Rev1 and Polη protein abundance is subjected to cell cycle regulation, being low at the G1 and G1/S stages, and peaking in the G2-phase. This is achieved mostly at the level of protein stability. However, in the case of Rev1, a cell cycle-dependent increase of its transcript levels was also observed [199,200,201]. It has been postulated that this pattern of regulation could be associated with the role and timing of Y-family polymerases in TLS, which would take place predominantly during G2/M in *S. cerevisiae* [199], supporting the concept that TLS polymerases act after bulk genomic replication. In agreement, Lopes et al. observed that TLS could counteract the accumulation of small ssDNA gaps in UV-irradiated yeast cells without affecting fork progression, which suggest that TLS occurs behind replication forks, in a post-replicative manner [202]. Conversely, Polη recruitment to DNA damaging sites occurs independently of the cell cycle stage in mammal cells, which suggests the ability of the polymerase to function in all cell cycle phases [203].

### 3.2. Rad5-Mediated Error-Free DDT Bypass Pathway

#### 3.2.1. Polyubiquitinated PCNA by Rad5- Error-Free Pathway

Monoubiquitinated PCNA may be modified by the heterodimeric E2 ubiquitin conjugase enzymes Ubc13-Mms2 and the E3 ubiquitin ligase Rad5, in *S. cerevisiae*, (or the Rad5 orthologous SHPRH and HLTF in humans). This modification involves K63-polyubiquitin chain extension onto K164 of PCNA. Genetic studies support this notion, since *RAD5*, *UBC13* or *MMS2* mutant cells are impaired in PCNA polyubiquitin chain formation, without affecting its monoubiquitination in vivo [163]. Polyubiquitinated-PCNA presumably signals error-free DDT pathway activation, mostly mediated by transient template switching, TS, in which the stalled nascent DNA strand uses the newly synthesized, undamaged strand of the sister chromatid as a template for replication [204,205] (see Section 3.2.3). Accordingly, mutants in this error-free pathway exhibit higher sensitivity to DNA damaging agents than mutants in the TLS pathway [206,207]. Polyubiquitin chain studies reveal that, rather than the total number of ubiquitin moieties, chain geometry is critical for error-free DDT bypass, suggesting that a still unknown receptor, with high selectivity for UBD, mediates TS activation [208]. Further work is required to uncover the complexity of ubiquitin as a signalling factor, the mechanisms by which polyubiquitinated-PCNA activates TS, and the effectors that are involved.

#### 3.2.2. Structural Features of Rad5, Interactions and Associated Activities

Rad5 has structured domains separated by IDRs. These domains include HIP116, Rad5p, the N-terminal (HIRAN) domain, helicase domain, and a RING domain, which is strikingly embedded into the helicase domain (Figure 3e). The HIRAN domain of Rad5 contains an oligonucleotide-binding fold (OB) that specifically binds the 3′ end of ssDNA, but prevents the binding of dsDNA binding [209,210,211,212].

Rad5 belongs to the SF2 superfamily of helicases [213]. Its helicase domain encompasses the half C-terminus of Rad5, and harbors seven conserved motifs, including Walker A and Walker B ATP-binding motifs. Furthermore, the helicase domain also binds DNA. The DNA-dependent ATPase activity of Rad5 becomes stimulated by either ssDNA or dsDNA. The yeast Rad5 DNA helicase activity is specialized in RF regression [212] (see Section 3.2.4). The Rad5 RING consists of a C3HC4 zinc finger-type domain formed by seven cysteine residues and one histidine residue coordinating two zinc ions [212]. It binds to Ubc13-Mms2 and is involved in Rad5 ubiquitin ligase activity [214]. Ubiquitin ligase and ATPase activities are essential for a functional error-free DDT pathway. Hence, mutations in any of the associated domains show sensitivity to DNA damage and increased mutagenesis, similar to *∆mms2* or *∆ubc13* mutant cells [215]. Remarkably, both ATPase and helicase domains overlap with the E3 ligase activity (see Section 3.2.3) through the polyubiquitination of PCNA, making it difficult to determine whether the helicase domain plays a catalytic role during DNA damage bypass. Through in vivo and in vitro characterization of helicase domain-specific mutants, Toth et al. show that the Walker B motif of the helicase domain is not necessary for Rad5-Ubc13 interaction, and that the Rad5 RING and helicase domains can function independently of each other [216].

In addition to Ubc13-Mms2, Rad5 physically interacts with Rad18, PCNA, and Rev1. Rad5 interacts with Rad18 through its N-terminus, where the HIRAN is located [214]. Although the functional implications of this association are unclear, the interaction between Rad5 and Rad18 may play a role in recruiting Rad5-Ubc13-Mms2 to monoubiquitinated PCNA. PCNA-binding to Rad5 is formed by its N-terminus, which harbors a PIP-like motif that also binds Rev1 [3,217]. Interestingly, Rad5 binds to unmodified PCNA and monoubiquitinated PCNA with analogous affinities [218]. Regarding Rad5-Rev1 interaction, crystal structural studies have determined that the Rev1 CTD binds specifically to a region of Rad5 containing a PIP-like motif [217]. This interaction promotes the recruitment of Polζ for TLS, pointing out that Rad5 may regulate Rev1-mediated TLS, thus playing a critical role in selecting TLS or error-free DDT pathways [217,219]. Nevertheless, the detailed function of Rad5 in TLS remains to be eluded. In addition, recent studies highlight a role for Rad5 in allowing for the bypass of both ssDNA gaps and methyl methanesulfonate (MMS)-induced DNA damage [220,221]. In the presence of MMS, Rad5 accumulates and forms nuclear foci during the S phase [222]. Several pieces of evidence show that a specific DNA lesion structure is required for the recruitment of Rad5 to the damaged site. Moreover, Polη or mutations in the BER pathway may impede or decrease Rad5 foci formation, respectively, supporting a possible role of Rad5 in mediating the pathway selection [220].

#### 3.2.3. Template Switch (TS) Model

The TS model for the error-free mechanism of DDT requires a process of strand invasion, which progresses in an HR-dependent manner [175]. Hence, some components playing a role in the HR also participate in the TS pathway. Frequently, specific HR intermediate structures are generated during the process. In summary, the TS begins with a strand invasion, in which the undamaged sister chromatid is transiently used as a replication template, exchanging the template for the blocked nascent strand, in order to carry over replication. This step is likely mediated by Rad51, and once the region containing the DNA lesion in the parental strand is replicated, the nascent strand switches back again to its original proper strand, leading to the restart of normal replication. Consequently, the appearance of derivative intermediates as X-shaped DNA or Rec-X structures (also called “sister chromatid junctions”, SCJ) occurs with some frequency throughout the process. Evidence based mostly on genetic approaches and the characterization of DNA intermediates using two-dimensional gel electrophoresis shows that these structures are usually resolved by the RecQ-helicase complex Sgs1/Top3/Rmi1 (BLM-TOPIIIa-RMI1-RMI2 in humans) [204,223,224]. Indeed, *∆sgs1* mutants accumulate X-shaped DNA structures at damaged RFs, without impairing fork progression, in a Rad51 and Rad52-dependent process [224,225]. Additionally, the endonuclease Mus81-Mms4 presumably cleaves these intermediates [226,227].

Chromatin remodeling changes that occur during replication seem to play an important regulatory role in promoting error-free DDT by TS, thus preventing mutagenic bypass and toxic recombination. Specifically, the member of the High Mobility Group (HMG) family, Hmo1, facilitates TS due to its ability to mediate DNA bending. At least two different Hmo1-mediated actions lead to the achievement of TS selection, namely the formation of SCJ and prevention of the SR pathway [228] (see Section 3.4.3).

#### 3.2.4. Fork Reversal Model

Fork reversal or fork regression is a regulated process used to stabilize stalled RFs and promote error-free lesion bypass, preventing ssDNA extension. This process requires the action of helicases and DNA translocases (reviewed in [229,230]). To overcome or facilitate the repair of a lesion that stalls the fork, nascent daughter strands dissociate from parental strands and anneal with each other, while the fork regresses and parental strands are reannealed, generating a four-way junction structure named “chicken foot” [231], where free ends on the reversed daughter strands must be protected from degradation. Fork reversal may imply TS, when the regressed lagging strand is used as the template to copy the leading strand.

In human cells, fork reversal has been described as a general mechanism for RF protection under mild genotoxic treatments [232]. Regarding fungi, this process has been mainly studied in *S. cerevisiae*, in checkpoint mutant cells under DNA damage conditions [231], or in Polα mutant cells, proficient for bulky DNA synthesis but deficient in re-priming [233]. In fact, fork reversal in yeast was considered a pathological structure that appears in checkpoint-mutant cells and was associated with the inability to restart RFs [234]. However, fork reversal has also been detected in wild type cells treated with camptothecin (CPT), a Topoisomerase I inhibitor, causing DNA torsional stress [235,236]. Therefore, it is assumed that fork reversal in yeast is not a general mechanism to protect stalled forks as in mammalian cells, but it may be enhanced when re-priming is not efficient. It might be a mechanism to pause and protect the stalled fork in the presence of DNA torsional stress, and/or a backup pathway to TLS or TS [237].

Different helicases and DNA translocases can trigger fork reversal. In vitro studies using DNA model molecules described that Rad5 is able to bind the 3′OH free end of the leading strand, through the HIRAN domain, and unwind the leading arm of the fork to trigger branch migration on the reversed fork [238,239]. However, a recent study questioned the ability of the HIRAN domain of *S. cerevisiae* and *Kluyveromyces lactis* Rad5 to bind ssDNA-3′OH, suggesting a different contribution to fork reversal [240].

Fork reversal is also triggered by Mph1 in *S. cerevisiae* and Fml1 in *Schizosaccharomyces*
*pombe* [241,242], both of which are orthologues of FANCM. Related to this, the Mph1 function is necessary to protect forks stalled by interstrand cross-link (ICL) lesions [243]. The Mph1 function in fork regression is positively regulated by Mhf1, Mhf2 and Mte1 [244], and it is negatively regulated by Smc5, a subunit of the Smc5/6 complex [245], necessary for the restart of stalled RFs, among other functions in maintaining genome stability (reviewed in [246]). Additionally, the inhibition of Rrm3 and Pif1 helicases by the checkpoint kinase Rad53 limits fork regression under replication stress in budding yeast [247]. Moreover, Exo1 nuclease represses fork reversal in budding yeast, probably by resecting the nascent regressed strand [248].

In mammalian cells, PCNA polyubiquitination mediated by UBC13 and ZRANB3 binding trigger fork reversal [249,250], although other DNA translocases are recruited to stalled forks through interaction with different factors (reviewed in [229,230]). In yeast, fork regression has been detected in wild type cells treated with CPT, but PCNA polyubiquitination has not been described under this condition [251]. This result suggests that PCNA polyubiquitination might not be required, at least for fork reversal initiation in yeast, although the protective role of Mph1 in cross-link-stalled forks in *S. cerevisiae* requires the ubiquitination of Pol30 [243]. Therefore, PCNA ubiquitination may be present during fork reversal under different damage conditions.

### 3.3. Alternative Ubiquitination Sites in PCNA

Alternative ubiquitination sites have been identified in *S. cerevisiae* (reviewed in [252]). K107 is specifically ubiquitinated in response to deficient DNA ligase I activity and to the accumulation of unligated Okazaki fragments [253,254]. This modification has been proposed as a DNA nick sensor. The ubiquitination of this alternative site is required to initiate the S phase checkpoint and promote a cell cycle delay when the maturation of Okazaki fragments is impaired. It depends on Rad5 (or Rad8 in fission yeast), together with the E2 partner formed by Mms2 and Ubc4, but not by Ubc13 [253,255]. K107 in yeast PCNA is positioned at the interface between PCNA subunits [255], suggesting that ubiquitination at this site might change the PCNA structure and the interaction between subunits, which would impair the correct function of PCNA. Related to this, in fission yeast, K107 ubiquitination was proposed to contribute to increased non-allelic crossovers, leading to gross chromosomal rearrangements (GCRs) depending on Rad52 [255].

In *S. cerevisiae*, PCNA is also ubiquitinated at K242 in response to defects in the maturation of Okazaki fragments [256]. This modification on K242 is related to a higher mutation rate, depending on TLS.

### 3.4. PCNA Sumoylation: Regulation of Homologous Recombination (HR)

#### 3.4.1. Srs2 Helicase Negatively Regulates the HR Pathway

Yeast PCNA is also conjugated with SUMO. Its significance, however, is less understood than ubiquitination. It is mostly accepted that the SUMO-modified PCNA leads to the suppression of HR through the recruitment of the Srs2 helicase [257], which removes Rad51 nucleoprotein filaments from ssDNA [258], and is critical to antagonize HR and to remove unproductive recombination intermediates [259]. The *SRS2* gene was originally identified in screens for suppressors of yeast *∆rad6* sensitivities to trimethoprim and UV [260]. The Srs2 function prevents HR, since suppressions mediated by an *∆srs2* mutant require functional components of the HR [261].

Srs2 activity involved in disrupting Rad51 nucleofilaments was termed “strippase” activity, which differs from the helicase function, although both entail the Srs2 translocase activity. Srs2 interacts with both Rad51 and SUMO-modified PCNA to complete its anti-recombination role during DNA replication. While the helicase domain is located at its N-terminus, Srs2 presents a flexible C-terminal region responsible for different protein interactions [262]. Accordingly, a conserved SIM and a degenerated PCNA interaction motif (PIM-like) are present at the very end of the C-terminus [184,263] (Figure 3g). Both motifs are required for optimal binding to SUMO-modified PCNA, which targets Srs2 to stalled RFs to suppress HR. Although Srs2 physically interacts with unmodified PCNA, the affinity of the interaction is significantly increased in SUMO-modified PCNA [264]. Alternatively, it has been proposed that the binding of Srs2 to SUMO-modified PCNA dissociates the replicative Polδ and the TLS Polη from the repair synthesis machinery and, thus, prevents the extension of recombination intermediates. To impede recombination, this latter mechanism involves only Srs2 recruitment through the SIM motif but not its translocase activity nor its Rad51 interaction [265].

Srs2 is evolutionarily conserved among fungal species. Interestingly, several species do not share the canonical C-terminus-containing motifs, as in *S. cerevisiae* [266]. In contrast, other species such as *S. pombe* present an additional related anti-recombinase protein [267]. Although Srs2 has additional roles during cell replication to warrant accurate genomic duplication, this review focuses on its functions as anti-recombinase.

#### 3.4.2. PCNA Sumoylation by Ubc9-Siz1

SUMO attachment to PCNA occurs primarily at the same K164 residue involved in monoubiquitination, and it is mediated by the E2 SUMO conjugase Ubc9 and the E3 SUMO ligase Siz1. To a minor extent, sumoylation at K127 has also been reported, in which only Ubc9 is required. K164 SUMO-modified PCNA occurs constitutively during the S phase in *S. cerevisiae*, but it is not related to cell cycle checkpoints. Despite sharing K164 residues, the levels of both SUMO and ubiquitin modifications do not seem to antagonize each other, since in *∆rad18* mutants, which are unable to ubiquitinate PCNA, the SUMO-modified PCNA levels remain invariable [163].

Ubc9 was isolated using SUMO affinity chromatography [261,268]. Siz1 is a member of the Siz/PIAS RING family of SUMO E3 ligases. Structural studies of Siz1 revealed that it contains an N-terminal PINIT domain, a central zinc-containing RING-like, SP-RING domain, and a CTD, termed SP-CTD (Figure 3f). Biochemical studies show that both the SP-RING and SP-CTD are required for the activation of the E2~SUMO thioester, while the PINIT domain is essential in interactions with the K164-PCNA [215].

Recently identified structural models of PCNA covalently modified by ubiquitin and SUMO indicate structural differences between them. Hence, ubiquitin has segmental flexibility and occupies discrete positions on PCNA. Conversely, SUMO associates by simple tethering and adopts extended flexible conformations. These differences point to distinct roles in DNA damage response regarding pathway regulation, and interacting proteins. Accordingly, this SUMO-PCNA structural model couples with Srs2 activities [269], since Srs2 requires a considerable degree of flexibility to execute its functions at the appropriate location. In addition, the SUMO moiety on PCNA-SUMO binds Rad18 and positions it to allow for the ubiquitination of K164 in other PCNA subunits of the trimer [270]. Similarly, Rad18 must exert its functions in various positions around the PCNA ring, hence, an expanded spatial range is also essential [270].

#### 3.4.3. Salvage Recombination (SR) Pathway

The salvage pathway, or salvage recombination (SR), is an alternative mechanism to DDT (reviewed in [271]). It is considered the last option, occurring at late S or G_2_ phases, since recombination events during replication must be highly controlled to avoid the accumulation of toxic recombination intermediates and genomic instability.

In budding yeast, during replication, sumoylated PCNA recruits the Srs2 helicase, which inhibits unscheduled recombination at ongoing RFs by disrupting Rad51 filaments [257,264]. The Srs2 anti-recombinogenic function is locally counteracted by Esc2 at stalled RFs [272], to allow for the error-free bypass of DNA lesions depending on recombination. Esc2 contributes to this pathway with the following two different functions: (i) it facilitates Elg1 association to damaged forks, enhancing PCNA unloading, together with bound Srs2, therefore, it limits the quantity of Srs2 specifically at damaged forks. (ii) In addition, Esc2 interacts through its SUMO-like domain (SLD), with the SIMs of Srs2 and Slx5, a subunit of the Slx5/Slx8 SUMO-targeted ubiquitin ligase (STUbL), causing local ubiquitination and proteasome-dependent degradation of Srs2. As a consequence, a low presence of Srs2 facilitates the Rad5-dependent TS pathway at stalled forks.

*∆rad5* cells are hypersensitive to various genotoxic agents, but the deletion of *MGS1*, which encodes a DNA-dependent AAA+ ATPase involved in maintaining genome stability [273], suppresses the sensibility of *∆rad5* mutants to MMS and HU. This result indicates the existence of an alternative repair pathway, inhibited by Mgs1 to prevent harmful recombination at stalled forks [274]. Mgs1 (WRNIP in humans) binds to PCNA and polyubiquitinated PCNA, has ssDNA annealing activity and interacts with Polδ, which may alter PCNA and Polδ interaction (reviewed in [275]). A lack of Mgs1 (or its ATPase activity) allows for this alternative bypass of DNA damage, implying the unloading of PCNA and Srs2 from stalled forks. This event facilitates the recruitment of Rad51 and the recovery of the fork by recombination, depending on Rad52, Rad59 and Polδ [274]. The recovery of RF also depends on Sgs1 to dissolve the recombination intermediates [276]. Similarly, the deletion of *SRS2* or expression of a Pol30 mutant version, that cannot be sumoylated (K164R, K127R), allows for SR in budding yeast [276], as Srs2 recruitment by PCNA sumoylation on either K164 or K127 inhibits recombination.

Although the different mechanisms for tolerating DNA damage are explained as different pathways, there may be a linear transition from one to another in vivo. How cells decide which pathway to use is still not well understood. Different factors and conditions such as the presence of replication stress or DNA damage, the type of DNA lesion, or the presence of topological stress, among others, might influence the selection of the DDT pathway that best fits each cell requirement.

### 3.5. PCNA Inner Surface Acetylation in Response to DNA Damage

The evolutionary highly conserved PCNA inner surface plays an important role for DNA polymerase processivity during replication and repair [3]. However, the dynamic interaction between DNA and the positively-charged sliding surface is not well understood [277]. Billon et al. showed that lysines on the inner surface of PCNA become acetylated in response to DNA damage, making cells more resistant to DNA-damaging agents. Specifically, K20 and K77 act as specific responders, since cell sensitivity to DNA damaging agents increases when they are replaced by acetyl-mimic glutamine residues. EcoI cohesin acetyltransferase acetylates K20 in vitro and in vivo in response to DNA damage, which stimulates repair by sister-chromatid-mediated HR. Moreover, the crystal structure of the PCNA ring acetylated on K20 reveals structural differences at the interface between PCNA subunits, which may suggest that transient conformational changes of the PCNA ring could have an effect on the sliding motion on the DNA. The effect of K77 acetylation has yet to be determined [278].

## 4. Comparative Study of PCNA from Systemic Pathogenic Fungi

The PCNA structure is conserved among fungi; however, certain differences exist in their sequences. Whether and how this variability influences the structure and regulation of PCNA must be experimentally confirmed. There is abundant information about the post-translational modifications that regulate PCNA function in humans and budding yeast [279], but less is known about other fungal species. In this review, the PCNA protein sequence from *S. cerevisiae* is compared with PCNA proteins belonging to 14 systemic pathogens, yeasts and filamentous fungi, from different *phylum*, namely *Ascomycota, Basidiomycota* and *Mucoromycota* (Figure 4). We focused on both higher and lower conserved domains or residues to understand regulatory modifications and structural features specific to fungi, in order to identify fungal PCNA as a therapeutic target.

Protein sequence alignment of PCNA from the contrasted fungal species indicates that the C-globular domain is less conserved than the N-globular domain, mainly in the region corresponding to the previously described P-loop or back loop (residues 183 to 195 in *S. cerevisiae*), where fungal PCNAs contain a 3_10_ helix, which is not present in human PCNA [269,280]. Besides, PCNA from *Cryptococcus neoformans* contains an insertion in this region of over 80 residues, absent in the rest of the species. IDCL and CTD sequences are well conserved, together with the G178 residue, which is involved in the stability of the ring, and along with K168, is required for Rad5 interaction [281]. K168 is conserved in all the compared fungal species, except in *C. neoformans*, where a threonine takes its place. The phosphorylation of tyrosine114 and 211 (Y114, Y211) stabilizes human PCNA, avoiding its ubiquitination and degradation by the proteasome, or alters PCNA interaction with mismatch repair (MMR) proteins, suppressing MMR, thus increasing mutational rates [282,283,284]. These modifications might be possible in fungal species as both residues, Y114 and Y211, are conserved, although they have not been reported to date. Interestingly, Pol30 expression with a Y114F mutation is unable to support budding yeast growth; therefore, this residue is essential for PCNA functions [281].

Some differences are detected in residues that are post-translationally modified to regulate the PCNA function in S. cerevisiae [252]. K164, which is ubiquitinated and sumoylated, is central to DDT regulation. Moreover, in budding yeast, Pol30 is also sumoylated at K127, although to a minor extent [163]. Although K164 is conserved in all the 15 compared species, K127, located in the IDCL sequence, is not; only S. cerevisiae and Candida glabrata contain a lysine at position 127. Nevertheless, the surrounding residues L126 and I128, required to correctly interact with Polδ, are always conserved. This modification on K127 is important to inhibit recombination events on forks, and to decrease Eco1 interaction with PCNA, which may impair cohesion establishment, mediated by the Eco1-dependent acetylation of cohesin (reviewed in [279]). Conversely, the acetylation of PCNA on K20 by Eco1 is related to recombination repair, as it diminishes replicative polymerase processivity and promotes HR [278]. This modification is conserved in budding yeast and humans. It has not been described in other fungal species, although K20 is conserved in all the species compared in this review.

Focusing on alternative ubiquitination sites on yeast PCNA, K107 and K242, occurring as a response to a deficiency in Ligase I activity or in the presence of unligated Okazaki fragments ([254,255,257]), and as observed in Figure 4, we can conclude that these positions are not well conserved among fungi. There is a very low conservation of the K242 position, although some of the fungal species in Figure 4 present a lysine at position 240, as human PCNA. K107 is located in the previously described J-loop [280]. This position is conserved in *S. cerevisiae*, *C. glabrata* and *C. neoformans*. However, in some other species there is a lysine residue at position 106 (*Candida albicans*, *Rhizopus microsporus*, *Mucor circinelloides*, *Lichtheimia ramosa* and *Lichtheimia corymbifera*), although post-translational modifications at this position have not been reported to date. In human cells, the down-regulation of DNA ligase I (LIG1A) also induces PCNA monoubiquitination [253]. In this case, the alternative ubiquitinated lysine is not known, since the K107 position is not conserved in human PCNA, although it contains a close-by lysine at position 110 [255], which is also present in other metazoans. Position 110 has a more conserved composition in all the fungal species compared in Figure 4, with arginine or lysine residues at this position. Curiously, all the compared species that contain a lysine at position 106, also present a lysine at position 110. Both, K107 in yeast PCNA, and K110 in human PCNA, are placed at the interface between PCNA subunits [255], suggesting that ubiquitination at these sites could alter their interaction and promote the release of PCNA from DNA. Nevertheless, the methylation of K110 in human PCNA was described as mediating trimerization and stabilizing the interaction of Polδ [285].

Differences between human and fungal PCNAs in post-translational modifications and target residues might be exploited to identify new treatments against fungal infections. Nevertheless, low conservation among fungi and/or coincidences in residues that are also modified in human PCNA would make this kind of approach difficult.

Mutations on PCNA that separately affect replication or repair, or mutations disturbing both functions have already been studied in budding yeast. For some of them, the PCNA structure and activity of wild type and mutant proteins have been compared [281,286]. Importantly, these studies concluded that in many cases, PCNA may incorporate mutations and still maintain its function in replication, although in some cases a higher mutation rate has been detected.

A comparison of different fungal species revealed interspecies incompatibility in the PCNA binding and coevolution of PCNA-partner interactions. Based on the IDCL sequence, two main groups were identified (group I and group II). Partners in species classified in group I are not able to bind chimeric constructs of PCNA with IDCL sequences present in group II, and partners from group II do not bind PCNA-containing group I IDCL sequences [287]. Hence, *S. cerevisiae*, *C. glabrata* and *Candida albicans* are classified in group I, and the rest of the compared species in Figure 4, in group II.

Recently, the structural resolution of PCNA from *C. albicans*, *Aspergillus fumigatus* and *Neurospora crassa* demonstrated interesting results about evolutionary differences/similarities in the PCNA structure and sequence requirements for binding PIP domain-containing interactors [280,288,289,290]. Differences in IDCL and the previously described J-loop, located between residues 105 and 110 in *S. cerevisiae* (Figure 4), implied structural variations that tend to limit PCNA interaction with partners from different species [280,290]. Even within group I, slight sequence variances could impair cross-species full complementation, as detected in *C. albicans* and *S. cerevisiae* [280]. These studies propose the following approach to control the replication of fungal pathogens: the use of model PIP-box peptides, with sequence variations that may enhance binding to fungal PCNA. Former experiments in the protein-directed evolution on PCNA, in order to increase the affinity for different partners, caused severe defects in replication and repair [291]. These results suggested that evolution has not favored strong PCNA-partner affinities. This might be in accordance with the many different functions that PCNA coordinates and the partner exchange that is required to perform them [292].

In fact, differences in PCNA domains necessary for interaction with various partners imply interspecies barriers that open the possibility of treating infections in humans, thus affecting the PCNA function specifically in pathogens [289]. However, a defective response to DNA damage in fungal pathogens has been associated with the appearance of drug-resistant mutants. Although defects in DNA damage repair cause a higher death rate, they also enhance the accumulation of mutations, which may be a driving force for microevolution. Some fungal pathogens present deficiencies either in repair genes or in signalling DNA damage [293,294,295,296]. Accordingly, genetic instability and the development of drug resistance have been reported under stress conditions or during host infection [297]. Moreover, different punctual mutations on PCNA have been identified to promote its function in replication, but were found to cause higher mutation and recombination rates [286]. Therefore, a profound comprehension of the function and structure of PCNA from different species and the significance of distinct punctual mutations is essential in order to develop specific and secure therapeutic targets against PCNA to treat mycotic infections.

## 5. Future Perspectives

Over the years, researchers have focused on unravelling DDT mechanisms. However, a deeper comprehension of its regulation is essential and will need to be addressed in the near future, including a detailed understanding of its molecular mediators. The activation of an accurate DDT molecular response is promoted by post-translational modifications of the PCNA sliding clamp. Thus, PCNA monoubiquitination drives the error-prone TLS pathway, whereas polyubiquitination enables error-free TS. PCNA may also be sumoylated and lead to Srs2 helicase recruitment, which negatively controls the SR alternative repair mechanism. Since TLS polymerases exhibit low fidelity, their error-prone activity must be tightly regulated and restricted to the vicinity of bulky DNA lesions. Nevertheless, the molecular events that balance between ending once the PCNA is monoubiquitinated or, on the contrary, continue to add a polyubiquitinated chain are still unknown. Concerning TLS, biochemical and structural analyses of TLS polymerases have allowed for considerable progress toward their characterization. However, critical questions remain to be solved, such as those related to PCNA conformational changes when different TLS polymerases bind to it, either cooperatively or competitively, and whether these changes affect polymerase switching. An interesting remaining issue is the way that other processes also related to replication contribute to the modification of DDT, in coordination with PCNA. In this sense, chromatin state and chromosome structure as well as to replication timing are important topics for future research. Moreover, quantitative proteomics studies might shed light on the identification of new PCNA-associated DDT modulators. Remarkably, another significant topic to be explored is the coexistence of various post-translational modifications in the same PCNA trimer, and how they interact with each other to determine the repair mechanism required in each condition.

Unfortunately, fungi are among the leading causes of opportunistic infections affecting immunocompromised patients. The high incidence of invasive mycoses in patients with HIV/AIDS represents an increasing threat to public health and underscores the urgent need for novel drug development strategies [298]. The variability in the PCNA sequence and structure between human and fungal pathogens opens the possibility to use specific drugs against PCNA functions in pathogens, impairing replication and growth. Research in fungi has revealed different sequence requirements for the interaction of PCNA with specific partners, which are not only highly helpful to the development of new anti-fungal treatments, but also crucial to understanding PCNA function in all organisms, including humans. This knowledge may be useful to find specific treatments against cancerous cell growth, based on PCNA inhibition. Nonetheless, the development of antifungal drugs is extremely difficult, since fungi, as eukaryotic organisms, share many similarities with human host cells. Therefore, targeting novel fungal PCNA epitopes may hinder the appearance of multiresistant species, highlighting the importance of exhaustively testing putative new fungal targets.

## Figures and Tables

**Figure 1 jof-08-00621-f001:**
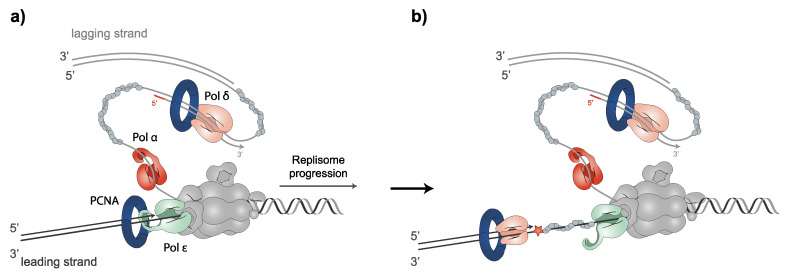
The polymerase switch facilitates the bypass of DNA lesions on the leading strand. (**a**) During unperturbed replisome progression, leading and lagging strands are synthesized by Polε and Polα/Polδ, respectively. (**b**) When the replisome encounters a DNA lesion that Polε is not able to tolerate, Polδ plays a key role in the initiation of leading-strand synthesis. The Cdc45-Mcm2-7-GINS (CMG) helicase complex is depicted in grey. Blocking DNA lesion is depicted as a red star.

**Figure 2 jof-08-00621-f002:**
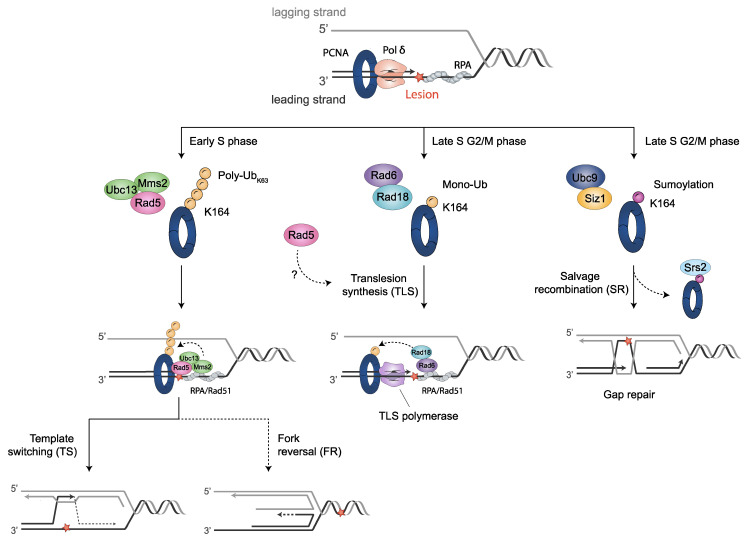
PCNA post-translational modifications regulate DDT pathways. When fork stalling persists, cells activate DDT mechanisms through post-translational modifications on PCNA. Monoubiquitination of PCNA at K164 by Rad6-Rad18 promotes the switch from Polδ to translesion synthesis (TLS) polymerases for error-prone TLS damage bypass. K63 extended polyubiquitination of PCNA on K164 by Mms2/Ubc13-Rad5 for error-free damage bypass mediates template switching (TS). This modification might be also implied in fork protection by fork reversal. Unloading of SUMO-PCNA bound to Srs2 (sumoylated by Ubc9-Siz1) provides the salvage recombination pathway (SR) alternatively to TS or TLS, either at stalled replication forks (RFs) or, as shown, at gaps left behind RFs after re-initiation. Cell cycle stages where DDT processes predominantly take place in yeast are indicated. Blocking DNA lesions are depicted as a red star.

**Figure 3 jof-08-00621-f003:**
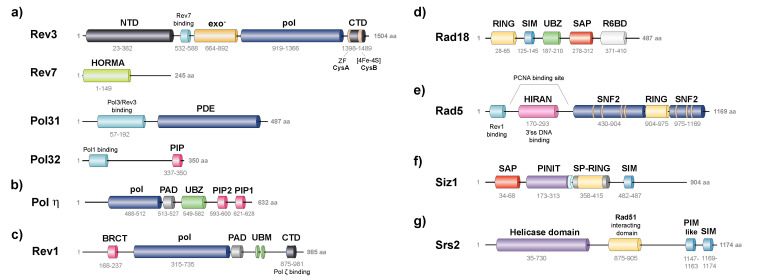
Structure organization of PCNA-interacting mediators involved in DDT in *S. cerevisiae*: TLS polymerases (**a**–**c**), modifying-E3 enzymes (**d**–**f**) and Srs2 helicase (**e**). (**a**) **Subunits of Polζ.** The catalytic subunit, **Rev3**, contains an inactive 3’-5’ exonuclease domain (exo^-^), a DNA polymerase domain (pol), and a conserved CysA and CysB sites in its C-terminal domain (CTD), containing a Zinc Finger (ZF) and a [4Fe-4S] cluster, respectively. The Rev7 binding site is located towards its N-terminal domain (NTD). **Rev7** contains the Hop1, Rev7 and Mad2 family domain (HORMA). **Pol31** contains a Rev3/Pol3 binding site and a phosphodiesterase domain (PDE). **Pol32** binds Pol1 and contains a PCNA-interactive motif (PIP). (**b**) **Pol****η** includes pol and PAD domains in its NTD, a ubiquitin-binding zinc finger motif (UBZ), and the PIP1 and PIP2 motifs at the CTD. (**c**) **Rev1** contains a pol domain, a polymerase associated domain (PAD), two small ubiquitin binding motifs (UBM), a small rev7 binding domain CTD and a BRCA1 NTD (BRCT). (**d**) E3 Ub-ligase **Rad18** contains a RING (Really Interesting New Gene) domain, the SUMO interacting motif (SIM), the UBZ motif, SAF-A/B, Acinus, Pias (SAP) domain, and Rad6-Binding Domain (R6BD). (**e**) E3 Ub-ligase **Rad5** contains the Rev1 binding domain, a HIRAN domain (HIP116 Rad5p N-terminal), the helicase domain (SNF2), and a RING domain. (**f**) E3 SUMO-ligase **Siz1** includes SAP, PINT and SP-RING domains, and the SIM motif. (**g**) **Srs2** helicase contains the helicase domain at its NTD, a Rad51 interacting domain, and a PIM and a SIM motif at CTD. The name and length (number of amino acids) of each PCNA-binding protein are indicated.

**Figure 4 jof-08-00621-f004:**
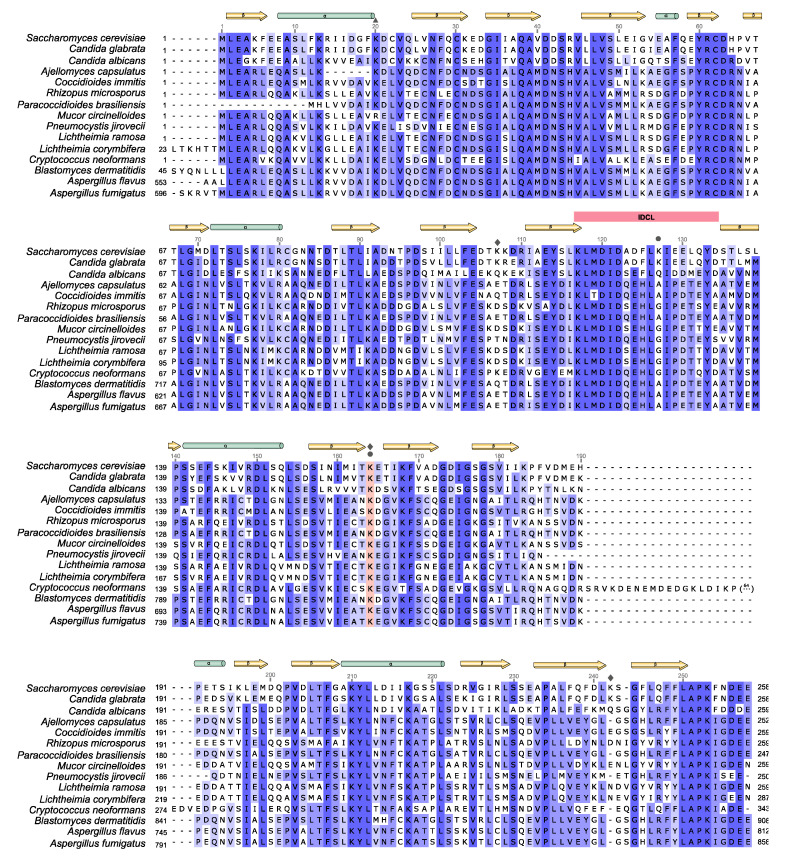
Alignment of multiple PCNA sequences of fungal pathogens, causing systemic infections, compared to *S. cerevisiae*. Multiple sequences alignment of *S. cerevisiae* PCNA and PCNA from 14 pathogenic fungal species is shown. *S. cerevisiae* (P15873), *Candida glabrata* (Q6FWA4), *Candida albicans* (Q5AMN0), *Ajellomyces capsulatus* (A6R5C7), *Coccidioides immitis* (A0A0J7B5C4), *Rhizopus microsporus* (A0A0A1P4Z3), *Paracoccidioides brasiliensis* (A0A1D2J4G1), *Mucor circinelloides* (S2K5N0), *Pneumocystis jirovecii* (A0A0W4ZHN6), *Lichtheimia ramosa* (A0A077WRZ8), *Lichtheimia corymbifera* (A0A068SE94), *Cryptococcus neoformans* (Q5K7Y2), *Blastomyces dermatitidis* (T5B6A2), *Aspergillus flavus* (B8N1A6), *Aspergillus fumigatus* (A0A0J5SJF1). PCNA sequences were obtained using Uniprot repository database (Uniprot entries are indicated in parentheses), and the sequence alignment was carried out using UGene software. Identical residues are shaded dark blue, whereas similar residues are shaded light blue. Secondary structural features are indicated above the sequences alignment, α-helices (yellow) and β-strands (green). Conserved IDCL motifs and K164 residues are shaded pink. The symbol on the upper part of the alignment indicates lysine modification—triangle for acetylation; rhombus for ubiquitylation; and circle for SUMOylation.

## Data Availability

Not applicable.

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
