# Peer review of "Post-Translational Modifications of PCNA: Guiding for the Best DNA Damage Tolerance Choice"

_jof, 2022, doi:10.3390/jof8060621_

Round 1

Reviewer 1 Report

This is a comprehensive review about DNA damage tolerance (DDT) pathways in yeast with some specific comparisons to mammalian cells. It is a good resource for yeast biologists, especially with the alignment of PCNA from various species. Overall, novelty in comparison to previous reviews is somewhat limited, and the authors omit several points that should be included.

Despite its long list of references, there are some important papers that have not been cited. The results of a genetic screen by the Boone laboratory showed that PCNA ubiquitination has a role during normal lagging strand synthesis (Becker et al. Plos Genetics, 2015). This was recently confirmed in mammalian cells (Thakar et al., 2020). The authors cite the mammalian work but not the yeast work, which is peculiar.

There are also alternative ubiquitination sites in budding yeast PCNA at K107 (triggered by ligase deficiency :Das-Bradoo et al, 2010, Nature Cell Biology; Nguyen et al., Plos ONE 2013) and K242, which supports translesion synthesis (Becker et al., NAR 2018). Alternative ubiquitination sites have also been reported in mammalian cells by Stephen Elledge's lab. Sites of mapped modifications in yeast should be included in the PCNA alignment (Figure 4).

Replication defects also cause a form of "defective-replication-induced mutagenesis" or DRIM, which was described by the Shcherbakova and Bielinsky laboratories. This has been left out for unknown reasons.

In Figure 1, the CMG helicase is not labeled, nor described in the figure legend.

In Figure 2, the depiction of the "salvage recombination" pathway is misleading, as it is not specifically triggered over a leason but utilized to fill replication gaps behind the fork.

Also, the authors should take into account and maybe stress in their review that Rev1 is only expressed in G2 in budding yeast, and not throughout S phase. TLS is therefore limited in S phase and TS is the preferred DDT pathway in yeast during S phase. This is not true in mammalian cells.

Some of the contents are misleading mostly due to language problems. Incorrect grammar leads to the distortion of sentences quite frequently throughout the manuscript, and the facts/findings are not as clearly stated as needed. An educated reader will guess the correct meaning, but someone new to the field will have difficulties understanding the text. The authors are not English native speakers, so this is not meant to be disrespectful, but it needs to be fixed and re-reviewed.

Reviewer 2 Report

PCNA is central to many DNA-related processes, like replication, repair, and checkpoints. It is also one of the main organizers of DNA damage tolerance (DDT) processes through its posttranslational modifications. The manuscript aims to summarize the current knowledge in this latter area. Given the contribution of DDT to genome stability, the subject is important and timely. However, for the same reason, review articles incorporating relevant new discoveries appear regularly.

Unfortunately, this manuscript in its current form does not meet the expectations. 1. It does not have a clear focus as its title would suggest. 2. Moreover, it leaves out several recent findings from the discussion. 3. Many cited references are misplaced making it difficult for the reader to navigate in the literature.

1.

More purposefulness could greatly improve the review. In this respect, chapter 2.1 (Loading and unloading PCNA onto duplex DNA) and chapter 2.2 (PCNA and replicative DNA polymerases: leading and lagging DNA) are unnecessary additions with too much detail that do not aid the understanding of the main subject. It would be enough to refer to relevant recent reviews of the field (eg. Ardel et al, Genes 12:1812, 2021, Acharya et al, Current Genetics 66:635- 2020). On the other hand, PCNA acetylation is mentioned only in one sentence, though as a DNA damage responsive PTM it would deserve more attention.

2.

Lane 65: the structure of PCNA on DNA and the sliding of PCNA on DNA are discussed, but the recent paper by Zheng et al, PNAS 117:30344-, 2020 is not mentioned. It solves the structure of PCNA on DNA with Pol delta, using yeast PCNA, while the cited, earlier papers used human PCNA with no polymerase.

Lane 331: rad8 delta strains are described as having no growth defect. In contrast, the poor growth of rad18 delta strains is well documented (eg. in SGD).

Lane 335: Rad18 Zn-finger is described to bind ubiquitin. However, according to two reports HLTF and SHPRH the human homologs of Rad5, and yeast Rad5 all bind to Rad18 Zn-finger (Zeman et al, JCB 206:183-, 2014; Frittmann et al, G3 11:jkab041 2021). Differing results should be included in the manuscript.

Lane 410: the Rad5 Walker B motif is described as supporting an ATPase activity and having a PCNA polyubiquitination structural scaffold-like role. Results of a recent paper by Toth et al, JMB 434:167437 2022, do not agree with the proposed role of the ATPase motif in PCNA polyubiquitination.

3.

The whole list of references needs to be carefully checked and corrected. I list just a few examples, but noticed many more:

Ref 228 is the same as ref.250

Lane 271: ref 154 refers to yeast pol eta, not human. Masutani, Nature, Johnson Science, 1999 are the correct ones

Lane 318: ref 176 is not the correct one. It is irrelevant.

Round 2

Reviewer 1 Report

The authors have responded well to the criticism and have made significant changes to their manuscript. This has added depth and clarity. The review has been much improved in both content and accessibility. The language editing and shortening of the text have helped overall readability.

Reviewer 2 Report

The authors addressed all my concerns.